# The Relationship between Carcass Condemnations and Tail Lesion in Swine Considering Different Production Systems and Tail Lengths

**DOI:** 10.3390/ani12080949

**Published:** 2022-04-07

**Authors:** Alice Gomes, Claudia Romeo, Sergio Ghidini, Madalena Vieira-Pinto

**Affiliations:** 1School of Agrarian and Veterinary Sciences (ECAV), University of Trás-os-Montes and Alto Douro (UTAD), 5001-801 Vila Real, Portugal; atc.gomes123@gmail.com; 2Department of Food and Drug, University of Parma, Via del Taglio 10, 43126 Parma, Italy; claudiarosa.romeo@izsler.it (C.R.); sergio.ghidini@unipr.it (S.G.); 3Istituto Zooprofilattico Sperimentale Della Lombardia e dell’Emilia Romagna, Via Bianchi 9, 25124 Brescia, Italy; 4Department of Veterinary Sciences, University of Trás-os-Montes and Alto Douro (UTAD), 5001-801 Vila Real, Portugal; 5CECAV—Veterinary and Animal Research Centre, University of Trás-os-Montes and Alto Douro (UTAD), 5001-801 Vila Real, Portugal; 6Associate Laboratory for Animal and Veterinary Sciences (AL4AnimalS), 5001-801 Vila Real, Portugal

**Keywords:** swine, meat inspection, tail biting, tail length, production system, post-mortem findings, carcass condemnations

## Abstract

**Simple Summary:**

Tail biting is considered a major welfare problem in swine production, associated with relevant financial costs for farmers. The European Union has reaffirmed the prohibition of tail docking practices, with all Member States establishing standards for the protection of pigs as well as measures to reduce the need for tail docking and tail biting prevalence. This research aims to assess the importance of tail lesions (using two different scores) and its influence on carcass condemnations considering different production systems and tail length. According to the results, higher tail lesion scores reflected higher total condemnations rates. An intimate association was encountered between the scarring score and total and local carcass condemnations. Tail length was also significant, with undocked animals presenting higher tail lesions scores. Organic batches reported more total condemnations. This research highlights the importance of tail lesions on carcass condemnations that may also be influenced by docking and type of production. These results suggest that scarred tail tissue should be included in the current tail surveillance program.

**Abstract:**

Tail biting has been recognised as an intractable problem in pig production. This study aims to evaluate tail lesion occurrence in slaughtered pigs and explore the relationship between carcass condemnations and tail lesion considering different production systems and tail lengths and to evaluate the importance of creating a detailed tail score classification that includes scarred lesions. Data on a total of 9189 pigs from 73 batches with different tail lengths (undocked; docked mid-length; fully docked) and from distinct production systems (conventional; conventional antibiotic-free and organic) were collected at a Spanish abattoir. Batches with higher tail lesion scores presented a significantly higher chance of total condemnation and total condemnation due to pyaemia, being even more associated with scarring score. The within-batches probability for local condemnations and local condemnation due to abscesses increased significantly with higher scarring scores. Regarding tail length, docked at mid-length and undocked carcasses presented significantly higher odds to be condemned due to abscess. Organic farms showed a higher probability of total condemnations. This research highlights the importance of tail lesions on carcass condemnations that may also be influenced by docking and type of production. Results suggest that scarring score should be included in the tail surveillance program.

## 1. Introduction

Harmful social behaviour, in particular tail biting, has been recognised as a common problem in pig production, being accountable for significant financial losses [1,2]. It is also considered a welfare issue since pigs suffering from tail injuries present pain, stress and frustration [3,4].

This behaviour has become more frequent over time as production intensified, and the environment became increasingly artificial. It is described as a multifactorial problem which is known to be triggered by a wide range of factors such as high stocking density, inadequate housing conditions, feeding-related issues, lack of environmental enrichment or health problems [4,5,6]. This behaviour is often seen in conventional indoor husbandry systems [2]. However, tail biting has also been documented in outdoor herds [7,8] and in organically raised pigs [9,10], indicating that it is not exclusive to the conventional husbandry system.

Tail biting can represent a problem during slaughter since it originates pathological findings which may imply total or local condemnations [2,11]. Abscesses, arthritis and signs of inflammation on hindlegs and front legs are more frequent in carcasses from tail bitten pigs [12,13]. Tail biting also leads to great economic losses for the farmers due to increased healthcare, additional animal management, and higher prevalence of carcass condemnation (either total or local) mostly related to abscessation [2,13,14]. In 1999, the UK alone registered a 4-million-euro expense related to tail biting [5]. According to [15], if the average prevalence of tail biting is at a level of 10%, the financial costs can be estimated at 2.3€ per slaughtered pig (which is approximately 1.6% of carcass value).

The European Union has stated its position regarding tail docking practices through the Directive 120/2008/EC and Recommendation (EU) 2016/336, which encourages all Member States to establish standards for the protection of pigs as well as measures to reduce the need for tail-docking and tail biting prevalence [16,17]. Although this procedure is prohibited by routine in Europe, many pigs are still exposed to docking to prevent tail damage later in life. The farmer mainly performs this procedure during the first week of the animal’s life without anaesthetics. If it is done later, it needs to be performed by a veterinarian with the administration of analgesia/anaesthesia to provide pain relief [17]. Tail docking cannot be done routinely, and it is only allowed if there is evidence of biting. It should be applied as last resort, and other measures related to environmental conditions, space allowance or enrichment material must be taken first [17]. According to [4] it is also necessary to consider that tail docking is a welfare problem in itself, since the procedure causes pain in piglets, can lead to the development of spinal abscesses, facilitates suboptimal production methods from a welfare point-of-view and does not extinguish the occurrence of tail biting. For that reason, it is imperative to consider the benefits and negative impacts of both docking and tail biting.

Based on several studies, Valros and Heinonen [18] reported that tail docking reduced the occurrence of severe lesions by half. In 2015, a study in Ireland where 99% of the pigs were docked still showed a 72.5% prevalence of tail damage along with a 2.5% incidence for severe lesions alone [19]. Two other Irish studies also showed that the frequency for severe tail lesions could be as high as 3.1% in docked animals [20], indicating that docking itself does not eliminate tail biting.

Bitten tails are frequently recorded at abattoirs during meat inspection in some countries (e.g., Norway, Sweden). The occurrence of tail lesions can be considered as an indicator of the pig’s welfare by reflecting housing conditions or animal management practices [21]. When questioned, pig producers acknowledged the potential of developing and applying meat inspection data as an animal health and welfare diagnostic tool [22]. However, abattoir data for tail biting are not very accurate and tend to underestimate tail damage [19,21]. A Danish study that included 111 herds showed that tail lesions, evaluated by clinical examination of animals on the farm, were actually double the number detected by meat inspection at the abattoir [23]. Hence, it is likely that meat inspection records at the abattoir detect only severe cases associated with ongoing infections and condemnations [24], creating a need to improve the tools for tail inspection. However, despite these limitations, recording tail scores at the abattoir during meat inspection may be considered a monitoring/surveillance cost-effective tool, functioning as an iceberg indicator for problems at farm-level [25].

This study aimed to: (i) evaluate the level of tail biting occurrence in slaughtered pigs, considering different tail lengths and production types; (ii) explore the relationship between carcass condemnations (either total or local) relating tail lesion evaluation, tail length and production type; (iii) assess the importance of creating a detailed tail score classification that includes scarred lesions.

## 2. Materials and Methods

From November 2020 to January 2021, data from meat inspection from 9189 pigs included in 73 batches were collected in a finishing pig abattoir located in the north of Spain. In this abattoir, animals from three different production systems are slaughtered, namely: conventional indoor, conventional indoor antibiotic-free and organic (Appendix A [26,27,28]).

### 2.1. Data Collection

Tail lesion and scarring scores were attributed to approximately one of every three pigs in the line, for a subset of 3636 pigs. For each of the examined batches (9189 pigs), the following information was recorded: farm identification number, type of production system, tail length (fully docked, docked at mid-length or undocked), number of animals in the batch, number and causes of total and local condemnations. Locally condemned parts included posterior thirds, anterior thirds, head, ribs, shoulders, hock, ham and rabada. Rabada is a cut commonly used in Spanish slaughterhouses and involves the intrapelvic part of the external obturator muscle, the medial ventral sacrocaudal muscle, the coccygeal muscle, sacrum and the tail. The veterinarian registered the condemned areas on official records. Regarding total condemnations, pyaemia was considered if the carcass presented purulent osteomyelitis or multiples abscesses, there was no distinction between these different conditions. In terms of local causes of carcass condemnation, all parts could be condemned due to the presence of inflammations, abscess, and purulent contamination. Ribs and anterior thirds could be condemned for pneumonia.

Abscess was only an eligible cause for local condemnation if it was found fully encapsulated and in a single area, with no signs of systemic infection, otherwise it was considered as a total condemnation. An anterior third could be rejected by pneumonia if there was an extensive or suppurative inflammation/infection associated with the pleura and/or lungs. Regarding purulent contamination, as the name suggests, the area was rejected not by the presence of the abscess itself but due to the leak of purulent content which defiled the area.

### 2.2. Tail Scores

Each tail was classified based on two different lesion scores: tail lesion and tail scarring. The first one was categorised as follows: (0) No evidence of tail biting; (1) Superficial lesions only, without the presence of blood; (2) Presence of puncturing wounds associated with tail bites, with possible presence of blood or inflammation; (3) Extended lesion associated with chewing with partial loss of tail tissue but with no loss of tail length; (4) Extended lesion associated with chewing with partial or total loss of tail length. (Figure 1).

Tail scarring was scored as follows: (C0) No scar; (C1) Visible scar with no tissue lost or alteration of tail length (mild scarring); (C2) Visible scar with presumable loss of tail length (severe scarring) (Figure 2).

The lesion score was adapted from Harley et al. paper [1]. The tail scarring score was based on a previous study developed by the authors. Based on the subset of animals examined and their assigned scores, for each batch, a batch-level tail lesion score and scarring score (defined hereafter as batch scores) were derived by applying the following equation ∑ (proportion of pigs with score_i_ × score_i_).

### 2.3. Statistical Analysis

The effect on batch scores of the production system (conventional, conventional antibiotic-free or organic) and the tail length (fully docked, docked at mid-length or undocked) were examined first through linear models. Pair-wise comparisons among significant variables were carried out through *t*-tests on differences of least square means (DLSMs), applying Tukey correction for multiple comparisons. Both batch-level scores were transformed (x^1/2^) prior to this analysis to achieve normality of residuals.

The probability for total condemnations was examined through two separate binomial logistic regressions using as response variables the number of total and the number of local condemnations on the total number of animals within the batch. Batch tail lesion scores and batch scarring scores showed only a weak correlation (Pearson’s *r* = 0.26), hence we included in these models both scores, and the production system and tail length as covariates. Similarly, the effect of the same explanatory variables on the probability of total condemnation due to pyaemia (e.g., the most frequent reason for total condemnation) of condemning specific parts of the carcass and of local condemnations due to abscesses was also explored.

In all the logit models, Firth’s penalised maximum likelihood estimation method was applied to account for quasi-separation of data and reduce rare events bias. Comparisons of significant variables with more than two levels were explored by means of Odds Ratio estimates (ORs) and their 95% Confidence Intervals (CIs).

All the analyses were carried out through PROC GLIMMIX and PROC LOGISTIC in SAS/STAT 9.4 software (Copyright © 2022, SAS Institute Inc., Cary, NC, USA).

## 3. Results

A total number of 73 pig batches were analysed in this study (Table 1). On average, each batch included 126 pigs (50–330). The most common production system was conventional (71.4%), followed by conventional antibiotic-free (16.4%) and organic production (12.2%). Carcasses with fully docked tails were the most frequent (78.4%), following the undocked (11.80%). Docked at mid-length carcasses were less frequent (9.85%) and were only observed in conventional and conventional antibiotic-free (Table 1).

### 3.1. Variation in Batch-Level Scores

Overall, batch-level scores had a mean value of 0.85 ± 0.03 for lesions and 0.17 ± 0.02 for scarring. Undocked animals presented a higher batch tail lesion and tail scarring score when compared to other tail lengths (1.10 ± 0,10 and 0.19 ± 0.04, respectively). Conventional production also demonstrated a higher batch tail lesion and tail scarring score when compared to other production systems (0.87 ± 0.04 and 0.18 ± 0.02, respectively). On the contrary, organically raised animals showed the lowest batch tail lesion score (0.75 ± 0.10) (Appendix A). Although the production system did not significantly affect tail scores, there was a tendency for conventional farms to display higher scarring scores than organic farms (*p* = 0.07).

Batch-level tail scores varied significantly with tail length (F_2,68_ = 6.72; *p* = 0.002), with undocked batches having significantly higher tail lesion scores compared to docked batches (DLSM: 0.18 ± 0.05; *p_adj_* = 0.001).

Regarding batch-level scarring scores, neither of the examined variables had an effect on scarring scores (both *p* > 0.05).

### 3.2. Relationship between Total Condemnations with Tail Scores, Production System and Tail Length 

The prevalence of total condemnations (*n* = 9189 animals) was 0.8% (*n* = 70), with 52.1% (*n* = 48 of 73) of the batches having at least one condemnation accounted for (Table 2). Pyaemia was the most common cause of condemnation in all productions, followed by peritonitis (0.5% and 0.1%, respectively). The only two condemnations due to Erysipelas were registered in organic production. Jaundice, organoleptic alterations of the carcass, inflammation and trauma existed only in conventional production in percentages lower than 0.1% (Table 2).

Batches with higher tail lesion scores presented a significantly higher probability of observing total condemnation (*p* = 0.0145, OR = 1.81; Table 3). Similarly, the probability of total condemnations in a batch was strongly associated (*p* = 0.0002) with tail scarring scores, with an increase of 0.5 units in the score leading to more than a 3-fold increase in the odds of having a total condemnation (Table 3).

The probability of observing total condemnations varied depending on the production system (*p* = 0.03), with organic farms showing a probability of total condemnation occurrence higher than conventional and conventional antibiotic-free productions (OR = 2.27 and OR = 4.36, respectively; Table 3). Indeed, 1.3% of organic batches registered at least one total condemnation in contrast to conventional and conventional antibiotic-free, who registered 0.8% and 0.3% respectively (Table 2).

The most frequent cause of total condemnation, pyaemia, was only influenced by tail lesions (*p* = 0.0126, OR = 2.06) and tail scarring (*p* = 0.0002, OR = 3.86), respectively (Appendix A). In highlight, batches with higher scarring scores had more than 3 times the odds of showing total condemnations due to pyaemia (Appendix A).

### 3.3. Relationship between Local Condemnations of Carcasses’ Anatomical Regions with Tail Scores, Production System and Tail Length

Regarding local condemnations, 692 out of 9189 (7.5%) pigs’ carcasses were locally condemned, with 94.5% (69/73) of the batches having at least one accounted (Table 4). In all production systems, ribs were the most condemned area (76.7%), followed by head (48%), anterior third (35.6%), rabada (31.5%), hock (23.3%), posterior third (16.4%), shoulder (2.7%) and ham (1.4%). The second most condemned region was the head (Table 4). Head condemnations, which included the neck region, were mainly related to the presence of abscesses.

Within-batches probability for local condemnations increased significantly with higher scarring scores (all *p* < 0.05, Table 5), while it was not affected by tail lesion scores. Ribs and rabada condemnations association with tail length was significant (*p* = 0.009 and *p* < 0.0001, respectively), with the former (ribs) showing higher odds of being condemned in fully docked than undocked pigs (OR = 1.85) and the latter (rabada) showing the highest odds in pigs with tail docked at mid-length when compared to fully docked or undocked (OR = 6.07 and OR = 3.84, respectively; Table 5).

Conventional pigs had a higher percentage of rejected parts when compared to the rest of the batches (7.9%). Condemnations of posterior thirds, hock, ham and shoulder were not observed in organic production (Table 4). Additionally, rabada was more likely to be condemned in organic systems compared to both conventional and conventional antibiotic-free (*p* = 0.0006; OR = 3.99 and OR = 2.97, respectively; Table 5).

Concerning specific condemnations due to abscesses, once again, they increased significantly with higher scarring scores (*p* < 0.0001, OR = 3.65; Appendix A). Tail length was also significant (*p* = 0.0002), with docked at mid-length and undocked carcasses having higher odds of showing abscess condemnations than fully docked carcasses (OR = 2.19 and OR = 1.70, respectively; Appendix A).

## 4. Discussion

We scored tail lesions and tail scarring at the slaughterhouse on pigs coming from different production systems and subjected to different tail-docking practices, to assess whether they are related to condemnations (either total or local). The importance of including scarred lesions on the tail surveillance programs was also evaluated.

Even though some animals in this study are observed with an intact tail (undocked) or with a longer tail (docked at mid-length), which refers to the progressive disuse of docking practice following the European Commission directive [17], it seems that its application to a totality of the animals is far from being achieved, even in production systems with lower animal density (e.g., organic production). Tail docking should not be performed as a routine procedure, only if there is clear evidence that other animals present ear or tail lesions [3]. According to the recent official audits carried out in the main European pig producing member states (Germany, the Netherlands, Italy, Spain and Denmark), from 2016 to 2018, the large majority of the animals were still being tail docked (95–100%) [29].

### 4.1. Variation in Batch-Level Scores

Our finding that undocked animals are more likely to develop severe lesions is consistent with previous studies that reported a reduction in tail biting behaviour following tail resection and would explain why undocked animals are prone to develop tail lesions [30,31,32,33,34,35]. The reason why tail biting incidence is lower when tail docking is performed is still not fully understood, it can be hypothesized that the tail may become less attractive as it is shorter and without long hairs at the tip [3].

According to [36], docking procedure is known to cause pain, discomfort, and distress to piglets who have the freedom to express their normal behaviour denied since, based on [37], the missing tail is a tool of communication and interaction among them. Thus, it is imperative to consider the benefits and negative impacts of tail docking, encouraging stakeholders to improve on-farm animal conditions and welfare in order to prevent tail biting, and therefore, the need to perform tail docking.

Other consequences of docking can include the risk of infections, mostly if it is performed under poor hygienic conditions [18]. The animal’s growth rate can also be affected by this procedure [36].

According to the results, there was a tendency for conventional farms to display higher scarring scores than organic farms. The advantages and disadvantages of conventional/organic systems concerning tail biting are controversial among researchers. Hansson et al. proved that conventionally raised pigs had a higher prevalence of tail lesions when compared to organic free-range pigs and that these findings were statistically significant [7]. However, recent studies indicate that organic free-range pigs had a higher prevalence of tail lesions when compared to conventional indoor [9,10]. Since the organically raised swine in this study are not entirely free-range (Appendix A) we cannot perform a direct comparison with previous studies.

The stocking density is a decisive factor in biting occurrence. There are several studies who describe a positive correlation between high stocking densities (100 kg/m^2^ or more) and tail biting prevalence [5,38]. One of the most typical behaviours for pigs is the exploratory, which was necessary to search for food in the natural environment, where the rooting allowed them to explore their surroundings [24]. When the floor is slatted or in concrete, as is the case for conventional production, the pigs tend to redirect this behaviour to other objects or animals [39]. This problem intensifies with slatted floors, which are used for their economic benefits, and have been positively associated with tail biting [5,39,40,41]. Considering that conventionally raised pigs presented in this research are housed in slatted floors with bedding and a stocked density of 0.65 m^2^ per head (Appendix A) this information can be corroborated into confirming the tendency for conventional production to have a higher tail biting prevalence.

### 4.2. Relationship between Total Condemnations with with Tail Scores, Production System and Tail Length

The rate of condemnation found in this study (0.8%) was similar to the one found by Vieira-Pinto et al. and Valros et al. [2,42]. In a recent study involving seven European countries (Denmark, Finland, Germany, Italy, Norway, Spain and Portugal), total condemnation rates ranged from 0.11–0.51%. This variation was most likely associated with the way the condemnation criteria are defined and being used relating the coding system of each country [43].

According to several studies, vertebral osteomyelitis and abscesses, defined as a form of pyaemia, are one of the most recurrent causes for post-mortem carcass condemnation [42,44]. According to a recent study that assessed the most common causes for total condemnations in seven European countries, osteomyelitis and multiple abscesses were always the leading pathological causes for carcass rejection. Peritonitis and jaundice were also indicated as the second most common causes in Norway and Spain, respectively [43].

Relating our data, since osteomyelitis and the presence of multiple abscesses was included in the same category (pyaemia) without distinction, we cannot perform a comparative analysis.

Since several studies support a close relation between tail biting and abscess formation or pyaemia, it was the only evaluated parameter by the statistical model [2,11,12,14,45]. Total condemnations due to pyaemia were influenced by both tails scores, with scarred lesions having the stronger effect. These results stress the importance of tail lesions as an important source of secondary infection leading to generalised disease such as pyaemia [12,46] and highlights the importance of using the scar lesion score during the classification of tails at the abattoir.

Two condemnations due to Erysipelas were registered and only in pigs from organic production. This result may be justified by the fact that the pigs from this type of production have exterior access and can be exposed to infected water or infected mammals’ urine or faeces (e.g., birds and rodents), which are a form of transmission for this disease [47].

The relationship between total condemnations and lesions scores highlights the financial impact of tail lesions due to condemnations rates, following the results presented by Valros et al. and Marques et al., who proved that higher tail lesion scores reflected higher odds for carcass condemnation [12,14].

In this research, besides evaluating tail lesions through a classical system, another classification system was included to assess the presence of healed lesions through scars. This was decided since, although tail lesions can be absent at the time of slaughter, it does not exclude the possibility that they have not occurred during the animal’s life. They could be already healed locally at the time of slaughter and therefore would not be detected during post-mortem inspection [12,48] or could even be hard to distinguish from docked tails during post-mortem evaluation [24]. In these cases, a scar was seen with or without a reduction in the size of the tail. The probability for total condemnations in a batch was strongly associated with tail scarring, which had a more significant impact than fresh lesions. Similarly, Valros et al. showed that healed tail damage also significantly increased the risk of condemnation [2]. This underlines scar evaluation as a valuable parameter to be included in any tail lesion score scheme used at the abattoir level.

From the different production systems, organic farms showed a higher probability of total condemnation. A Danish study comparing meat inspection on finishing pigs from indoor and free-range systems also proved that free-range systems had higher odds for presenting septicaemia at slaughter, which is a condition which implicates total condemnation [10]. In this study, fully antibiotic restrictions were applied to the organic system (Appendix A), which might justify why this production system has higher condemnation rates, as it was also underlined by Lis Alban et al. [9].

### 4.3. Relationship between Local Condemnations of Carcasses’ Anatomical Regions with Tail Scores, Production System and Tail Length

Regarding local condemnations, the prevalence observed is similar to the one reported in a study performed in Finland, where 7.0% of the observed pigs were locally condemned [2].

In all production systems, ribs were the most condemned area, followed by head, anterior third, rabada, hock, posterior third, shoulder and ham. Similar results were previously found by [2,19]. In this study, condemnations of ribs were related to pleurisy, where the adherence of the pleura made it impossible to detach it from the ribs. In a recent study mentioning partial condemnations causes in seven European countries, the most common cause for condemnation was attributed to pleuritis and pneumonia [43]. These results reflect the importance of respiratory diseases as a common finding in swine productions worldwide and its economic impact during meat inspection [49].

The second most condemned region was the head, which included the neck area. Since it is one of the most common inoculation areas for intramuscular and subcutaneous injections, it may be hypothesised that these rejections are associated with poor practices in this procedure [50]. Heads condemnations were also not associated with any of the examined variables, which supports that head abscesses are potentially related to incorrect practices rather than tail lesions [51].

The within-batch probability for local condemnations was only influenced by scarred lesions, highlighting the importance of scarring over fresh tail lesion scoring. This result is consistent with what was found by Valros et al. in Finland, which demonstrated that healed lesions, in combination with bite marks or bruises, were associated with partial carcass condemnations and abscesses [14].

Condemnations of posterior thirds, hock and shoulder were not observed in organic production. This result is in accordance with the ones presented by [9] showing that hock scarred lesions were more prone to occur in conventional production than in organic. Additionally, a Danish study comparing meat inspection on finishing pigs from indoor and free-range systems also proved that free-range systems presented a lower incidence of hoof abscess [10].

In terms of production system and its association with locally condemned parts, only rabada was influenced, with organic pigs displaying higher odds for rabada condemnations. To the best of our knowledge, no other studies evaluated this specific cut before, therefore a direct comparison cannot be performed. However, due to its anatomical location, it can be hypothesised that rabada condemnation may be related to tail abscessation. Alban et al. reported that tail lesion, tail infections and abscesses in hindquarters were all more frequent in organic production when compared to conventional, that difference being highly significant [9].

In general, local condemnations due to abscess were only influenced by scarred tail lesions. There results follow the conclusions which several studies had already established, which is the association between tail damage and abscessation [2,11,12,14,45].

In this research, carcass condemnations were always associated with tail damage, being scarred lesions the constant factor between total and local condemnations. These results stress that, once again, the significance of healed tail lesions and the importance of using the scar lesion score during the classification of tails at the abattoir.

## 5. Conclusions

This study evaluated the association of tail scores with production system, tail length, and carcass condemnations in pigs slaughtered in Northern Spain. Our results suggest that undocked pigs were more likely associated with severe tail lesions and abscess condemnations. Thus, the negative impact of the docking procedure on the pigs’ welfare should be systematically weighted under a risk analyses approach. As both tail scores increased, the probability of observing total condemnation in a batch was higher, the scarring score having a more substantial effect. When it was narrowed down to condemnations due to abscess regarding local condemnation, only the scarring score remained a constant indicator, with a more relevant role when compared to tail lesion score. This strongly supports the importance of developing more studies featuring tail scarring assessment in slaughterhouse meat inspection. Organic farms showed a higher probability for total condemnation, probably because these animals are deprived of antibiotics, and therefore are more prone to develop systemic infections, which leads to condemnations. Regarding tail condition, we can only affirm with certainty that organically raised pigs are not exempt from developing tail lesions.

It is well known that tail biting can lead to tremendous economic losses due to augmented condemnation rates. There is an emergent need for surveillance of this type of lesions both at the slaughterhouse and farm level. It would be highly beneficial to create a communication channel between these stakeholders, where the farm could have a better perception of the batch’s health and welfare based on meat inspection data. Using meat inspection data as a diagnostic tool could contribute to establish a positive relation and trust among stakeholders in the pig industry [22]. This research concludes that the tail scarring score presented a close relationship with total and local condemnations, showing that more studies should be performed in order to include scarred lesions in the tail surveillance program.

## Figures and Tables

**Figure 1 animals-12-00949-f001:**
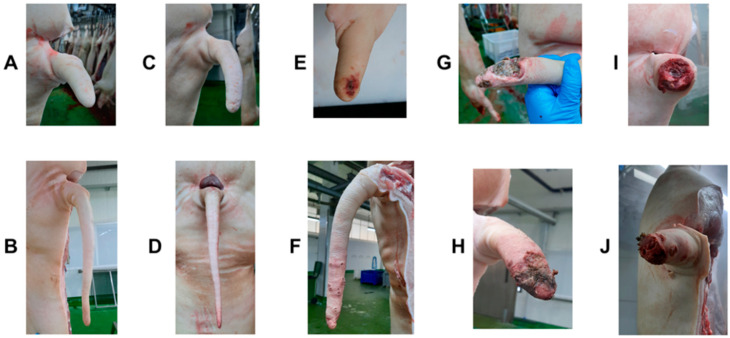
Example pictures of tails scored according to the tail lesion scoring system used (See Section 2.2 for details): (**A,B**) Score 0—No evidence of tail biting; (**C,D**) Score 1—Superficial lesions only, without the presence of blood; (**E,F**) Score 2—Presence of puncturing wounds associated with tail bites, with possible presence of blood or inflammation; (**G,H**) Score 3—Extended lesion associated with chewing with partial loss of tail tissue but with no loss of tail length; (**I,J**) Score 4—Extended lesion associated with chewing with loss of tail length.

**Figure 2 animals-12-00949-f002:**
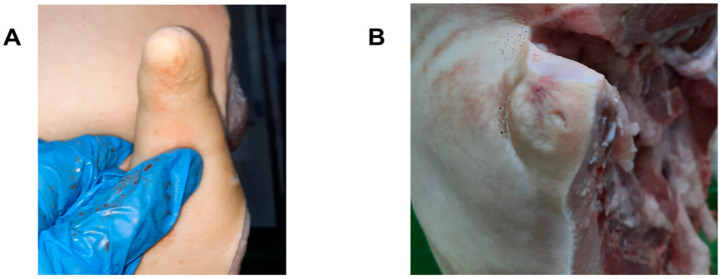
Example pictures of tails scored according to the tail scarring scoring system used (See Section 2.2 for details): (**A**) Score C1—Visible scar with no tissue lost or alteration of tail length (mild scarring); (**B**) Score C2—Visible scar with presumable loss of tail length (severe scarring).

**Table 1 animals-12-00949-t001:** Description of the total number of animals slaughtered (*n* = 9189) with respect to the number of animals, batch size, production system and tail docking.

	N	B	%
Slaughtered animals	9189	73	100
Animals examined at individual-level	3636	73	39.57
Production system			
Conventional	2596	51	71.40
Organic	443	10	12.18
Conventional AB ^1^	597	12	16.42
Tail docking			
Fully docked	2849	57	78.36
Conventional	2142	42	58.91
Organic	356	8	9.79
Conventional AB ^1^	351	7	9.65
Undocked	429	9	11.80
Conventional	194	4	5.34
Organic	87	2	2.39
Conventional AB ^1^	148	3	4.07
Docked mid-length	358	7	9.85
Conventional	260	5	7.15
Organic	0	0	0
Conventional AB ^1^	98	2	2.70

^1^ Antibiotic-free; N—total number of animals; B—number of batches; %—percentage of total.

**Table 2 animals-12-00949-t002:** Batch-level (% of batches with at least one condemnation/no. of examined batches) and all population-level (% of pigs/no. of examined pigs) prevalence of total condemnations (and respective cause) and its distribution over the various production types. Unless otherwise specified, 95% confidence intervals (CI) of the prevalence are reported within brackets.

	Batch-Level (*n* = 73)	Animals Slaughtered (*n* = 9189)	Conventional (*n* = 7201)	Conventional AB ^2^ (*n* = 1348)	Organic (*n* = 640)
TC ^1^	52.1%, 48 (40.59–63.52)	0.8%, 70 (0.6–0.9)	0.8%, 58 (0.6–1.0)	0.3%, 4 (0.01–0.6)	1.3%, 8 (1–1.5)
Causes for total condemnations					
Pyaemia	38.4%, 28 (27.2–49.5)	0.5%, 49 (0.4–0.7)	0.6%, 42 (0.4–0.8)	0.2%, 3 (0–0.5)	0.6%, 4 (0.01–1.2)
Peritonitis	13.7%, 10 (5.81–21.6)	0.1%, 10 (0.04–0.2)	0.1%, 7 (0.03–0.2)	0.1%, 1 (0–0.2)	0.3%, 2 (0–0.7)
Jaundice	2.7%, 2 (0–6.5)	0.02%, 2 (0–0.05)	0.03%, 2 (0–0.07)	0	0
Organoleptic alterations	4.1%, 3 (0–8.7)	0.03%, 3 (0–0.07)	0.04%, 3 (0–0.1)	0	0
Inflammation	4.1%, 3 (0–8.7)	0.03%, 3 (0–0.07)	0.04%, 3 (0–0.1)	0	0
Trauma	1.4%, 1 (0–4.0)	0.01%, 1 (0–0.03)	0.01%, 1 (0–0.04)	0	0
Erysipelas	1.4%, 1 (0–4.0)	0.02%, 2 (0–0.05)	0	0	0.3%, 2 (0–0.7)

^1^ Total condemnations; ^2^ Antibiotic-free.

**Table 3 animals-12-00949-t003:** Logistic regression model exploring batch-level variation in the occurrence of total condemnations in pigs’ batches (*n* = 73) at the slaughterhouse. For significant variables, odds ratio estimates (OR) and their 95% confident intervals (CI) are presented, with estimates for continuous scores calculated for a 0.5 unit increase. Significant *p*-values and ORs are highlighted in bold.

Response Variable	Explanatory Variable	Statistic	*p*-Value		Odds Ratio	
					Estimate	95%CI
TC ^1^	Batch tail lesion score	Χ^2^_1_ = 5.98	**0.0145**		**1.81**	**1.12–2.91**
	Batch scarring score	Χ ^2^_1_ = 13.81	**0.0002**		**3.24**	**1.74–6.02**
	Production system	Χ ^2^_2_ = 7.27	**0.0263**	Organic vs. Conventional	**2.27**	**1.07–4.81**
				Organic vs. Conventional AB ^2^	**4.36**	**1.38–13.7**
				Conventional AB ^2^ vs. Conventional	**0.52**	**0.19–1.40**
	Tail length	Χ ^2^_2_ = 0.06	0.97			

^1^ Total condemnations; ^2^ Antibiotic-free.

**Table 4 animals-12-00949-t004:** Batch-level (% of batches with at least one condemnation/no. of examined batches) and all population-level (% of pigs/no. of examined pigs) prevalence of local condemnations and its distribution over the various production types. Unless otherwise specified, 95% confidence intervals (CI) of the prevalence are reported within brackets.

	Batch-Level (*n* = 73)	Animals Slaughtered (*n* = 9189)	Conventional (*n* = 7201)	Conventional AB ^2^ (*n* = 1348)	Organic (*n* = 640)
LC ^1^	94.5%, 69 (89.3–99.8)	7.5%, 692 (7.0–8.1)	7.9%, 565 (7.2–8.5)	7.5%, 48 (5.5–9.5)	5.9%, 79 (4.6–7.1)
Parts Condemned					
Anterior third	35.6%, 26 (24.6–46.6)	0.7%, 62 (0.5–0.8)	0.8%, 56 (0.6–1.0)	0.2%, 1 (0–0.5)	0.4%, 5 (0.05–0.7)
Posterior third	16.4%, 12 (7.9–24.9)	0.15%, 14 (0.1–0.2)	0.2%, 13 (0.1–0.3)	0	0.1%, 1 (0–0.2)
Head	48%, 35 (36.5–59.4)	0.5%, 48 (0.4–0.7)	0.5%, 39 (0.4–0.7)	0.5%, 3 (0–1.0)	0.5%, 6 (0.1–0.8)
Ribs	76.7%, 56 (67.0–86.4)	4.9%, 450 (4.5–5.3)	5.2%, 375 (4.7–5.7)	5.2%, 33 (3.4–6.9)	3.1%, 42 (2.2–4.04)
Rabada	31.5%, 23 (20.9–42.2)	0.9%, 84 (0.7–1.1)	0.8%, 59 (0.6–1.0)	1.8%, 11 (0.7–2.7)	1.04%, 14 (0.5–1.6)
Hock	23.3%, 17 (13.6–33)	0.3%, 28 (0.2–0.4)	0.3%, 21 (0.2–0.4)	0	0.5%, 7 (0.1–0.9)
Shoulder	2.7%, 2 (0–6.5)	0.02%, 2 (0–0.05)	0.01%, 1 (0–0.04)	0	0.1%, 1 (0–0.2)
Ham	1.4%, 1 (0–4.0)	0.01%, 1 (0–0.03)	0.01%, 1 (0–0.04)	0	0

^1^ Local condemnations; ^2^ Antibiotic-free.

**Table 5 animals-12-00949-t005:** Logistic regression model exploring batch-level variation in local condemnation probability and parts condemned within pigs’ batches (*n* = 73) at the slaughterhouse. For significant variables, odds ratio estimates (OR) and their 95% confidence interval (CI) are presented, with estimates for continuous scores calculated for a 0.5-unit increase. Significant *p*-values and ORs are highlighted in bold.

Response Variable	Explanatory Variable	Statistic	*p*-Value		Odds Ratio	
					Estimate	95%CI
LC ^1^	Batch tail lesion score	χ^2^_1_ = 1.33	0.50			
	Batch scarring score	χ^2^_1_ = 57.7	**<0.0001**		**6.28**	**3.9–10.09**
	Production system	χ^2^_2_ = 3.22	0.20			
	Tail length	χ^2^_2_ = 4.07	0.13			
Anterior third	Batch tail lesion score	χ^2^_1_ = 1.33	0.25			
	Batch scarring score	χ^2^_1_ = 4.54	**0.033**		**2.13**	**1.06–4.26**
	Production system	χ^2^_2_ = 3.21	0.20			
	Tail length	χ^2^_2_ = 1.29	0.52			
Head	Batch tail lesion score	χ^2^_1_ = 0.15	0.69			
	Batch scarring score	χ^2^_1_ = 1.95	0.16			
	Production system	χ^2^_2_ = 0.57	0.75			
	Tail length	χ^2^_2_ = 4.16	0.12			
Ribs	Batch tail lesion score	χ^2^_1_ = 1.19	0.28			
	Batch scarring score	χ^2^_1_ = 26.3	**<0.0001**		**2.18**	**1.59–2.84**
	Production system	χ^2^_2_ = 4.04	0.13			
	Tail length	χ^2^_2_ = 9.44	**0.0089**	Fully docked vs. Undocked	**1.85**	**0.36–0.83**
				Undocked vs. Docked at mid-length	0.72	0.43–1.20
				Docked at mid-length vs. Docked	0.76	0.53–1.10
	Batch tail lesion score	χ^2^_1_ = 0.13	0.72			
	Batch scarring score	χ^2^_1_ = 40.29	**<0.0001**		**7.61**	**4.07–14.25**
	Production system	χ^2^_2_ = 15.0	**0.0006**	Organic vs. Conventional	**3.99**	**1.98–8.04**
Rabada				Organic vs. Conventional AB^2^	**2.97**	**1.32–6.67**
				Conventional AB ^2^ vs. Conventional	1.34	0.72–2.48
	Tail length	χ^2^_2_ = 44.47	**<0.0001**	Undocked vs. Fully docked	1.56	0.77–3.13
				Docked at mid-length vs. Undocked	**3.84**	**0.12–0.55**
				Docked at mid-length vs. Fully docked	**6.07**	**3.57–10.33**

^1^ Local condemnations. ^2^ Antibiotic-free.

## Data Availability

The data presented in this study are available on request from the corresponding author. The data are not publicly available due to privacy reasons.

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
