# Peer review of "The Relationship between Carcass Condemnations and Tail Lesion in Swine Considering Different Production Systems and Tail Lengths"

_animals, 2022, doi:10.3390/ani12080949_

Round 1

Reviewer 1 Report

The manuscript is very interesting and provide several new information on the relationship between carcass post mortem findings and tail lesion.

The authors should improve the title of the manuscript and the tables for a better understanding.

The reference along the manuscript need to be revised following author instruction.

Clarify the number of observation

Please see the attached PDF file

Author Response

Dear Reviewer,

First, I would like to express my gratitude for all the considerations regarding this paper. The input was appreciated. Please consider the attached word file with all revisions made (this included the suggestion and comments other reviewers have also proposed). Supplementary materials were also revised. The “Track Changes” option was used so it would be easier to follow. The in-text references were all corrected following the authors instruction guideline.  Please let me know if you consider any changes in this final revision to be insufficient or if there is any aspect to be further improved.

I would like to address some specific comments individually:

L153 and 156: Please add the figure after the description

            R: Done.

Table 3 and Table 5 (former Table 6): I suggest to better organize the table for a better understanding

            R: I appreciate your input. However, I am having difficulties in optimizing this table. I would kindly ask you to enlighten me with your detailed opinion on this. Could you maybe refer a table example I could lean on?

Thank you once again.

Sincerely,

Alice

Reviewer 2 Report

The paper describes the prevalence at a single abattoir of tail lesions of different severity attributed to tail biting and relates this to the prevalence of carcass condemnations for different causes. It also seeks to evaluate the influence of production system and tail (docking) length on these condemnations. The information is very important for the understanding of the economic (and welfare) consequences of tail biting, and this paper makes a valuable contribution to knowledge. Although the overall sample size is high, I have some concerns about whether the number of individual batches investigated was really adequate to dissociate the highly confounded effects of tail length and production system, and would welcome clear information on this batch breakdown in the sample size Table and some consideration of this in the discussion. The proposal for an abattoir surveillance system including scarring score is an important one and is strongly supported by the data presented.

Specific comments:

L26. I don’t think it is an emerging problem as it has been apparent for many years. Maybe an ‘intractable’ problem.

L27, L100. I think it was tail lesion occurrence that was evaluated

L33, L204. Significantly higher

L55. exclusive to

L59. What is ‘member inflammation’?

L62. These data are now rather old. It might be better to cite more recent information. See references in  e.g. Niemi et al. Front Vet Sci (2021) 8: 682330.

L128. Delete one of the repeated ‘only’

L138. ‘perforation’ of what? There must be some perforation to cause a lesion.

L159. Was it the absolute number or the proportion/probability that was analysed? [OK, later I see this was used as a binary Y/N indicator]

L182. Give also the standard errors for these batch means.

L228, L233, L238. What is Table F? should this be Table 6?

L283. evaluated

L300. Giving reference to a whole book is not very helpful. Could the specific chapter within this book be referenced instead?

L346. According to Table 2, the Erysipelas condemnations occurred the ‘conventional without AM’ system.

L371. Is it tail biting or tail lesions that these studies reported?

L376. It is the condemnations, not the tail lesions which they linked to a prohibition on AMs. This text needs revising.

L383. How can it be that organic animals are not ‘free-range’? This term needs to be defined more precisely. Do you mean that they had outdoor access but were not at pasture?

L394. ‘detach’

L399. Maybe need to specify if this are includes the neck, since this is the actual injection site.

L412. What is meant by ‘scar’ here?

L443. I miss in the discussion and conclusions any statements about the extent to which the higher condemnation in organic systems is due to the much higher prevalence of undocked tails in this system (therefore increased risk of tail lesion) or to other aspects of the system. What do your data indicate regarding this important question? Did the data have sufficient power (no of batches) to dissociate this confound?

Table S1. According to EU Regulations, Organic pigs cannot be weaned at < 40 days of age. Teeth reduction should be prohibited except under specific exceptions. Correct the spelling of ‘exceptions’ in this table.

Fig 1. The score 4 definition needs to include loss of tail length

Table 1. This should include the number of batches for each category, as well as the number of animals, since tail biting is often a batch-level phenomenon.

Table 3. No bold text is apparent in my copy (as suggested by legend)

Table S3. Is this again the odds of showing at least one condemnation for this cause? Please make legend clear.

Table 5. The order of showing data on % and number of batches/pigs is reversed compared to Table 2, and not in accord with the legend. Please standardise this (% followed by number) to avoid confusion.

Author Response

Dear Reviewer,

First, I would like to express my gratitude for all the considerations regarding this paper. The input was appreciated. Please consider the word file with all revisions made (this included the suggestion and comments other reviewers have also proposed). Supplementary materials were also revised. The option “Track Changes” was used so it would be easier to follow. Please let me know if you consider any changes in this final revision to be insufficient or if there is any aspect to be further improved.

I will address the specific comments individually:

L26. I don’t think it is an emerging problem as it has been apparent for many years. Maybe an ‘intractable’ problem.

R: Suggestion accepted.

L27, L100. I think it was tail lesion occurrence that was evaluated

            R: Altered from “tail biting” to “tail lesions”.

L33, former L204 (now L230). Significantly higher

            R: Done.

Former L55 (now L65). Exclusive to

            R: Altered

Former L59 (now L69). What is ‘member inflammation’?

            R: Altered from “member inflammation” to “signs of inflammation on hindlegs and front legs”.

Former L62 (now L72 and 73). These data are now rather old. It might be better to cite more recent information. See references in e.g. Niemi et al. Front Vet Sci (2021) 8: 682330.

            R: Thank you for the suggestion. I added valuable info from the suggested article relating the financial costs implied by higher prevalence of tail biting.

Former L128 (now L140). Delete one of the repeated ‘only’

            R: Done.

Former L138 (now L50). ‘perforation’ of what? There must be some perforation to cause a lesion.

            R: I removed that term since it wasn’t correct. I only intend to make the reader understand that tail lesions classified as grade 1 were “scratch” lesions, without the presence of blood and exuberant inflammation.

L159. Was it the absolute number or the proportion/probability that was analysed? [OK, later I see this was used as a binary Y/N indicator]

Former L182 (now L201). Give also the standard errors for these batch means.

            R: Added.

Former L228, L233, L238 (now L254, L259, L264). What is Table F? should this be Table 6?

            R: Altered from “Table F” to “Table 5” (former Table 6). I was a number ahead in the table count.

Former L283 (now L362). evaluated

            R: Altered.

Former L300 (now L377-379). Giving reference to a whole book is not very helpful. Could the specific chapter within this book be referenced instead?

            R: This reference was removed. The EFSA report cited alone provided a good reference.

Former L346 (now L433). According to Table 2, the Erysipelas condemnations occurred the ‘conventional without AM’ system.

            R: Table 2 had a typo regarding the Erysipelas condemnations. Info regarding Conventional AB and organic were swapped in the data transcription. This is now corrected in the table.

Former L371 (now L437). Is it tail biting or tail lesions that these studies reported?

            R: Both of these studies evaluated carcasses’ tails and attributed them scores. On Marques et al. paper, they stated that carcass condemnations were associated with the presence of what the authors describe to be tail-biting lesions. On Valros et al. paper it was stated that higher levels of both whole and partial carcass condemnations were found in carcasses with tails with most lesion types.

Former L376 (now L457). It is the condemnations, not the tail lesions which they linked to a prohibition on AMs. This text needs revising.

            R: This line of though was misleading and confusing. I moved and altered this text fraction since it was misplaced. Please check also L389.

Former L383 (now L396). How can it be that organic animals are not ‘free- range’? This term needs to be defined more precisely. Do you mean that they had outdoor access but were not at pasture?

            R: Our organically raised swine had indoor facilities with permanent outdoor access but not pasture. In Supplementary materials Table S1 is indicated the space allowance for indoor/outdoor space for organics.

Former L394 (now 503). ‘detach’

            R: Done.

Former L399 (now 508). Maybe need to specify if this area includes the neck, since this is the actual injection site.

            R: Done.

Former L412 (now L520/521). What is meant by ‘scar’ here?

            R: It was described with this term in the cited paper. Altered from “scar/hock lesion” to “hock scarred lesion” for better understanding.

L443. I miss in the discussion and conclusions any statements about the extent to which the higher condemnation in organic systems is due to the much higher prevalence of undocked tails in this system (therefore increased risk of tail lesion) or to other aspects of the system. What do your data indicate regarding this important question? Did the data have sufficient power (no of batches) to dissociate this confound?

            R: I took the liberty to improve Table 1 with some important info which could be enlightening for this matter. If we look at the organic production data, there are few undocked tail swine. In our results, we also verified that organically raised animals showed the lowest batch tail lesion score, therefore, the higher total condemnations rates in this production may not be justified by tail lesions. When the logistic model was run, we concluded that although the production system did not significantly affect tail scores, there was a tendency for conventional farms to display higher scarring scores than organic farms. In order to properly address the matter, the sample size of undocked organically raised swine should have been larger.

Table S1. According to EU Regulations, Organic pigs cannot be weaned at < 40 days of age. Teeth reduction should be prohibited except under specific exceptions. Correct the spelling of ‘exceptions’ in this table.

            R: Corrected. Regarding age of weaning, in organic production “21-25 days” was replaced with “approximately 40 days”. Regarding teeth reduction, “grinding or partially cut only if necessary, within the first 3 days” was replaced with “prohibited except under specific exceptions”.

Fig 1. The score 4 definition needs to include loss of tail length

            R: Done.

Table 1. This should include the number of batches for each category, as well as the number of animals, since tail biting is often a batch-level phenomenon.

            R: Table 1 was altered and more info relating batches was provided.

Table 3. No bold text is apparent in my copy (as suggested by legend)

            R: Corrected.

Table S3. Is this again the odds of showing at least one condemnation for this cause? Please make legend clear.

            R: This is the first time these results are presented in the paper. Table 3 refers to total condemnations considering all causes (pyaemia, peritonitis, jaudice, trauma, etc). This table only refers to total condemnations due to pyaemia.

Table 5. The order of showing data on % and number of batches/pigs is reversed compared to Table 2, and not in accord with the legend. Please standardise this (% followed by number) to avoid confusion.

            R: Was former Table 5 and is now Table 4. This was also corrected to match Table 2.

Thank you once again.

Sincerely,

Alice